# Calcium Homeostasis, Transporters, and Blockers in Health and Diseases of the Cardiovascular System

**DOI:** 10.3390/ijms24108803

**Published:** 2023-05-15

**Authors:** Ghassan Bkaily, Danielle Jacques

**Affiliations:** Department of Immunology and Cell Biology, Faculty of Medicine and Health Sciences, Université de Sherbrooke, Sherbrooke, QC J1H 5N4, Canada

**Keywords:** calcium, calcium homeostasis, calcium overload, hypercalcemia, Cav1, Cav2, Cav3, L-type calcium channel, P/Q-type calcium channel, N-type calcium channel, R-type calcium channel, T-type calcium channel, calcium channel blockers, G protein, GPCR, calcium pumps, sodium/calcium exchanger, endoplasmic reticulum, nucleus, mitochondria

## Abstract

Calcium is a highly positively charged ionic species. It regulates all cell types’ functions and is an important second messenger that controls and triggers several mechanisms, including membrane stabilization, permeability, contraction, secretion, mitosis, intercellular communications, and in the activation of kinases and gene expression. Therefore, controlling calcium transport and its intracellular homeostasis in physiology leads to the healthy functioning of the biological system. However, abnormal extracellular and intracellular calcium homeostasis leads to cardiovascular, skeletal, immune, secretory diseases, and cancer. Therefore, the pharmacological control of calcium influx directly via calcium channels and exchangers and its outflow via calcium pumps and uptake by the ER/SR are crucial in treating calcium transport remodeling in pathology. Here, we mainly focused on selective calcium transporters and blockers in the cardiovascular system.

## 1. Introduction

Calcium (Ca^2+^) plays a vital role in development, aging, and death. This ion regulates most cellular functions via its broad role as a second messenger at the cytosolic and nuclear levels [1,2,3,4]. It is directly and indirectly implicated in life and death signals [5], sperm and ovulation maturation [6], fecundation [6], differentiation [5], contraction [7,8,9,10], secretion (endocytosis/exocytosis) [11], proliferation [12,13,14], and memory [15]. Its high extracellular concentration (near 2 mM compared to its intracellular concentration, near 50 nM) and its positive charges contribute to plasma membrane surface charges which affect plasma membrane fluidity. Its intracellular Ca^2+^ homeostasis is regulated by ionic transporters such as calcium channels, exchangers, and electrogenic pumps [16,17] (for summary, see Table 1 and Figure 1). These ionic transporters are also present on the membranes of intracellular organelles, such as the endoplasmic/sarcoplasmic reticulum, the mitochondria, and the nucleus [16,17]. In addition, calcium micro- and nanodomains are formed at the inner side of the mouth of a transporter that contributes to intracellular calcium release, uptake, and signaling [18,19,20,21,22]. These micro/nanodomains enable the cell to use calcium in a specific part of the cell [18]. These micro/nanodomains were given names (sparklet, spark, blink, syntilla, and puff) depending on the type of transporter that created such domains [18] (Figure 1). For example, sparklet originates from the calcium voltage-operated calcium channel (VOCC) opening, which creates a calcium microdomain by activating calcium-induced calcium release from the endoplasmic/sarcoplasmic reticulum (ER/SR) ryanodine type 2 receptor (RyR2). This latter type of microdomain calcium release is called spark (Figure 1). The different calcium micro/nanodomains contribute to several cell functions, such as excitation–contraction (EC) coupling, excitation–secretion (ES) coupling, and excitation–gene expression (EG) coupling.

The regulation of normal intracellular calcium homeostasis in all cell types, particularly highly differentiated cells such as those of the cardiovascular system, is crucial to proper cell function. The dysregulation of Ca^2+^ transport through the plasma and intracellular organelles, such as the ER/SR, the mitochondria, and the nucleus shown in Figure 1, contributes to calcium overload in the cytosol, the mitochondria, and the nucleus. This calcium overload contributes to morphological (by promoting cell hypertrophy) and functional remodeling (by dysregulating EC and EG couplings as well as cell survival) [5]. Thus, knowing the calcium transporters that regulate normal intracellular calcium homeostasis is vital in order to shed more light on the importance of calcium in health and disease.

This review is mainly focused on selective calcium transporters and blockers in the cardiovascular system. In addition, we will also briefly describe other types of non-selective calcium transporters (ROCCs) of different cell types, such as neurons.

## 2. Calcium Ionic Transporters

The name for calcium comes from the Latin name for lime, calx, as lime was first isolated from it as calcium oxide (CaO). This ion species has a high affinity for water. Thus, its hydration/dehydration in the outer and inner sides of the membrane plays an essential role in its mobility through the pores of different calcium channels, exchangers, and pumps [23]. As proteins, membrane phospholipids have a high-affinity binding site to calcium [23]. Thus, binding the double-positively charged calcium to extracellular sites of the membrane phospholipids highly contributes to the density of positive charges on the extracellular side of the plasma membrane [24]. This high density of positive extracellular charges on the plasma membrane contributes to determining the membrane potential, stability, fluidity, and functioning of voltage-operated ionic transporters, including VOCCs [25] (please see Table 1 and Figure 1). In addition, calcium influx through specific plasma membrane ionic transporters increases intracellular membrane surface charges and depolarizes the membrane potential [26]. The contributions of calcium to intracellular and extracellular membrane surface charges maintain resting intracellular calcium homeostasis. Changes in the normal cell resting membrane potential level, particularly in highly differentiated ones, such as cardiac, skeletal, and nerve cells, will affect normal EC, ES, and EG couplings.

### 2.1. Calcium Channels

In general, VOCCs are formed from a ring of four subunits. VOCCs are found in the membrane of excitable cells (for a review of the structure of VOCCs, please refer to [27]). Calcium-permeable channels are grouped into two major types: VOCCs, in the presence of extracellular calcium, are only selective for calcium, and receptor-operated calcium channels (ROCCs), also called ligand-gated channels, are not calcium selective in the presence of normal ionic milieu. VOCCs are sensitive to changes in membrane potential caused by electrical stimuli, whereas ROCCs are activated following the binding of their respective ligands [28,29,30,31,32,33,34,35,36,37,38,39].

There are different types of calcium-selective VOCCs (Figure 1). These differ in their threshold activation (Figure 2A and Figure 3A), voltage sensitivity (Figure 2A) to their opening probability (Figure 3B), conductance (Table 1), kinetics of activation and inactivation (Figure 2D), and their response to organic calcium channel blockers and activators [28,29,30,31,32,33,34,35,36,37,38,39] (please see Table 1). In addition, ROCCs are more challenging to define due to the difficulty of measuring their voltage sensitivity versus their ligand dependencies [36] and determining their single-channel levels of conductance due to the limitation of the method of recording used [40,41,42,43].

### 2.2. Classification, Function, and Pharmacology of VOCCs

As proposed in the literature [39,44,45,46,47], VOCCs are only selective for calcium in the presence of normal physiological conditions. They are classified into five major types: L- (high threshold or long-lasting) (Figure 2A,C), T- (low threshold or fast transient) (Figure 2A,B), N- (neither or neuronal), P/Q- (Purkinje neurons), and R-type (steady-state resting) (Figure 3) channels (see Table 1).

Ten isoforms of calcium-selective VOCCs exist. Among these are four subtypes of L-type channels (Cav 1.1, 1.2, 1.3, and 1.4) (Table 1 and Figure 1). These subtypes of channels (Cav 1.1 to 1.4) are the most studied and are the target of clinically used calcium antagonists [48]. Thus, our review will focus on this family of VOCCs. Different genes encode these four subtypes of L-type channels and are tissue-dependent (different subtypes present in various types of tissues) [27]. Similarly to other VOCCs, L-type calcium channels, once activated by voltage depolarization, undergo three states: resting, open, and closed. However, they have differences at the pharmacological and biophysical levels compared to other VOCCs (please see Table 1).

The L-type channel family (Cav 1.1 to 1.4) are all sensitive to three types of calcium antagonists: dihydropyridines (DHPs) (nifedipine, isradipine, nitrendipine, felodipine), phenylalanines (verapamil, D-600, D-888), and benzodiazepines (diltiazem) (for review, please refer to reference [48] and Table 1). All three types of blockers act on a specific channel state: either open (verapamil and diltiazem) [49] or closed (DHPs) [50]. In other words, their effects depend on the plasma membrane resting potential of the cell: −85 mV for cardiomyocytes (verapamil and diltiazem) and −65 to −55 mV for vascular smooth muscle (VSM) (DHPs) [50]. For this reason, DHPs act mainly at the level of VSM (the resting potential is more depolarized than that of cardiac muscle) and are primarily used as antihypertensive drugs [51]. Furthermore, blocking Ca^2+^ entry into the cell prevents activation of Ca^2+^ release from the ER/SR RyR [52]. The Cav 1.1 subtype is mainly present in skeletal and, to a lesser extent, in cardiac muscles (please see Table 1). In contrast, the Cav 1.2 subtype is primarily present at the VSM level and relatively less in cardiac and neuronal cells. On the other hand, Cav1.3 are present in the endocrine, cardiac, and neuronal cells, whereas Cav1.4 are primarily located in the retina (Table 1 summarizes this information). For this reason, a difference exists between regulating the vascular Cav1.2 calcium channel by cAMP (channel blocking) and that of the cardiac and neuronal Cav1.3 (channel stimulation). Although the L-type calcium channel is highly selective to calcium, it is only in the absence of extracellular calcium that the channel is more permeable to Ba^2+^ [53] and to several inorganic cations, such as strontium (Sr^2+^) and sodium (Na^+^) [27,28,54,55,56]. As shown in Figure 2 and Table 1, the L-type channel has a high activation threshold (−10 mV in 2 mM Ca^2+^) and conducts a slow current with a large unit conductance (25 pS in the presence of 110 Ba^2+^); the single channel conductance is considered to be the fingerprint of an ion channel [47] (Table 1). This type of channel is highly sensitive to DHPs. Therefore, it is called a DHP-sensitive calcium channel [48] compared to other types of VOCCs. It is also inhibited by other organic L-type calcium channel blockers such as verapamil and diltiazem [5,57,58,59] (please see Table 1). However, several divalent ions such as nickel (Ni^2+^), cadmium (Cd^2+^), cobalt (Co^2+^), and manganese (Mn^2+^) block the L-type calcium channels in the concentration range from 0.5 to 20 nM. Lanthanum (La^3+^) [60,61] (Table 1) can also be indirectly modulated by neurotransmitters, enzymes, and drugs such as angiotensin II, endothelin-1, neuropeptide Y, and beta-agonists [39,57,58,59,62,63]. It is modulated by second messengers such as cyclic AMP-dependent protein kinase (PKA [64,65]), cyclic GMP-dependent protein kinase (PKG) [66,67], protein kinase C (PKC), and calmodulin (CaM) [68]. An increase in intracellular calcium activates the two latter kinases. The blocking sequence of inorganic L-type calcium channel inhibitors in decreasing potency is La^3+^ > Co^2+^ > Mn^2+^ > Ni^2+^ > Mg^2+^ [60,61]. DHP-sensitive L-type channels are found in all excitable cells, such as neurons and muscle cells. Their voltage- and time-dependent kinetics facilitate the conversion of membrane depolarization into an intracellular calcium signal that drives a cellular response. Some examples of these phenomena include EC coupling in cardiac and smooth muscle cells, the activation of glycolytic metabolism in skeletal muscle, the ES coupling of endocrine and exocrine gland cells, and neurotransmitter release from peripheral neurons [63,69,70].

The T-type or fast transient Ca^2+^ channel (compared with the L channel) can be activated by small depolarizations (−70 mV, in 10 mM Ca^2+^) and has a relatively fast voltage-dependent inactivation; Figure 2 shows a typical example [53]. For the structural aspect of the channel, please refer to reference [27]. Three subtypes of T-type VOCCs were reported: Cav3.1 (present in neurons and heart cells), Cav3.2 (present in the heart and neuronal cells), and Cav3.3 (present in neurons); Table 1 summarizes this information [53]. Compared to L-type channels, the T-type channels are equally permeable to Ba^2+^ and Ca^2+^ [53]. Contrary to L-type channels, an increase in intracellular calcium does not regulate their inactivation gate [71,72,73,74]. This channel type is insensitive to DHPs, other organic calcium agonists, and antagonists [71,72,73,74]. It has a small unit conductance (8 pS in 110 mM Ba^2+^) (see Table 1) [46,75,76,77]. T-type calcium channels resemble the L-type calcium channel in that both are permeable to Na^+^ in the absence of divalent cations [75,76,77,78,79]. However, unlike the L-type Ca^2+^ channel, the T-type channel possesses the same unit conductance in the presence of Ca^2+^ and Ba^2+^ [46]. They are more resistant to blockade by Cd^2+^ ions than the L-type channel. Its blockage by omega-CgTX VIA is weak and reversible [80]. These channels are insensitive to the application of β-adrenergic receptor agonists but are highly sensitive to blockade by Ni^2+^ ions and amiloride (see Table 1) [81]. Similarly to L-type channels, T-type channels are found in neuronal, endocrine, exocrine, cardiac, smooth muscle, and skeletal cells (Table 1 summarizes this information) [77,82]. In addition, the cellular distribution and properties (activation at −70 mV) of T-type channels suggest that they may play a role in pacemaker current and rhythmic activity (see Table 1) [83]. T-type channels are inhibited by compounds, such as pimozide and penfluridol, from the diphenylbutylpiperidine family [5,70]. The diphenyldiperazine flunarizine can also block T-type channels, preferentially for Cav3.1 and Cav3.3 [53]. However, none of these blockers succeeded in being used in the clinic as an antagonist of T-type VOCCs.

The N-type calcium channel has an intermediate activation threshold between the T- and L-type channels (−30 mV in 10 mM Ca^2+^), a reasonably rapid inactivation, and an inactivation constant with an intermediate value compared with that measured for the T- and L-type channels [34,35,36,37]. The N-type calcium channel has a unit conductance of 13 pS in 110 mM Ba^2+^ [37]. It is sensitive to omega-conotoxin VIA (w-CgTX VIA) and to Cd^2+^ and is resistant to Ni^2+^ and DHPs [37] (Table 1). Flow measurements accompanied by electrical recordings from brain synaptosomes, sympathetic ganglions, and dorsal root neurons suggest that N-type channels have similar ion permeability to the L-type Ca^2+^ channel [36,37,70]. Therefore, their expression may be restricted to neuronal membranes. The pharmacological properties of N-type VOCCs are characteristic of a calcium entry pathway leading to neurotransmitter release in sympathetic neurons, nerve (motor) endings, and synaptosomes [84,85,86]. Two other VOCCs were reported in neurons, P- (Purkinje) and Q-types [84,87]. The single channel conductance of both types of channels is 10–20 pS (Table 1) (for review, please see ref. [88]). These types of calcium-selective VOCCs are mainly found in synaptic terminals, are high-voltage dependent, and are implicated in vesicle release and neurological diseases such as ataxia, migraine, and Alzheimer’s [84].

The steady-state voltage-dependent and G-protein coupled resting calcium channel (R-type) was reported to be present in many cell types, including those of the cardiovascular system [2,8,17,89,90,91,92]. This channel is only permeable to calcium and is insensitive to organic and inorganic L-type calcium channel blockers, except for isradipine (PN 200-110, Table 1) [92]. For more information, please see references [11,89,92]. This channel has a single-channel conductance of 24 pS (in the presence of a patch pipette containing 110 mM Ca^2+^ and 10^−5^ M nifedipine) [89], and Figure 3 shows an example. The R-type calcium channels play an important role in regulating resting calcium homeostasis. It is the only type of calcium channel present at the nuclear membrane and in both excitable and non-excitable cell types. It also plays an important role in cellular cardiovascular diseases since its frequency of opening as well as the time of opening increase by cardiovascular active factors such as insulin, ET-1, Ang II, NPY, TNF-α (tumor necrosis factor), and PAF (platelet-activating factor) [2,8,17,89,90,91,92].

**Figure 3 ijms-24-08803-f003:**
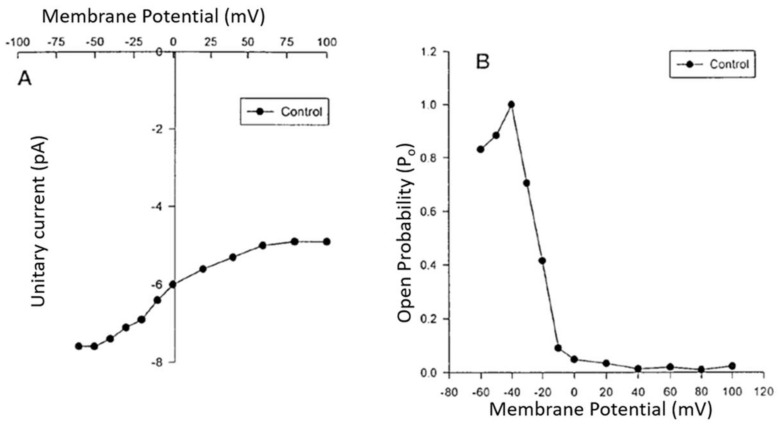
Current to voltage (I/V) relationship curve (**A**) and open probability/voltage relationship (**B**) of the voltage-dependent steady-state R-type Ca^2+^ channel in human aortic VSMCs recorded using the patch clamp technique. Modified from [11,92].

### 2.3. The Receptor-Operated Calcium Channels (ROCCs)

Receptor-operated channels (ROCs), also called transmitter-gated ion and ligand-gated channels, open directly in response to their respective ligands [93]. These are structurally formed by a ring of five subunits (or domains) [94], giving this type of channel a larger pore when compared to the VOCCs. In general, ROCs are not selective for a single type of ion [94]. An example of this type of channel is the nonspecific ion channel associated with the nicotinic acetylcholine receptor [95,96] and the Cl^−^ channels associated with GABA and glycine receptors [97].

Not all ROCs are permeable to Ca^2+^. In normal physiological ionic solution, those permeable to calcium and other ionic species are called receptor-operated Ca^2+^ channels (ROCCs). Several studies suggest that, similarly to VOCCs, there are different types of ROCCs [39], such as vasopressin-activated Ca^2+^ channels in smooth muscle cells and parathyroid hormone-activated Ca^2+^ channels in osteosarcoma cells [98,99], as well as transient receptor potential (TRP) mainly in the kidney [100], N-methyl-D-Aspartate receptors (NMDARs) [101,102], and α-amino-3-hydroxy-5-methyl-4-isoxazole propionic acid receptors (AMPARs) in the central nervous system [101] (Table 1). In this review, we focused on TRP1 and TRP2 because they were also reported to be present in the cardiovascular system [103], and their single-channel levels of conductance were reported [104,105,106,107,108]. We also briefly reviewed, for the same reasons mentioned previously, two other ROCCs, the NMDAR, and the AMPAR, because of the recent interest in them at the cardiovascular system level [102,109]. Contrary to VOCCs, the ROCCs, in addition to Ca^2+^, carry K^+^, Na^+^, Ca^2+^, Ba^2+^, and Mg^2+^ ions. Although some reports indicate that they are voltage-dependent, this aspect is still controversial [103,104,105,106,107,108,110,111,112,113,114,115,116,117,118,119,120]. Thus, they cannot be classified in the calcium-selective VOC category.

### 2.4. ROCCs: TRPC1, TRPP2, NMDARs, and AMPARs

The TRP channel superfamily includes 28 related non-selective anion channels that display different permeability degrees to calcium classified into six subfamilies: canonical (TRPC), vanilloid TRP (TRPV), melastatin TRP (TRPM), ankyrin TRP, mucolipin TRP, and polycystin TRP [121,122,123]. TRPC channels were the first to be cloned and are intensively studied to this day. Their single-channel levels of conductance, particularly TRPC1 and TRPPC2, were determined at the biophysical levels, which is not the case for most TRP subfamily channels [122]. These subfamilies are among the most abundant ion channels in the cardiovascular and nervous systems. TRPC1 and 2 (TRPP or PC2) channels have been well-studied in the last decade and have allowed us to describe some of their properties (Table 1) [103,104,105,106,107,108,110,111,112,113,114,115,116,117,118,119,120]. Single-channel recordings of native and expressed TRPC1 and TRPP2 in kidney insect SF9 and kidney HSG (or HEK-293) cell lines are very noisy and show an overlapping subconductance [107,113] which is difficult to differentiate between the different types and subtypes of TRP channels. For example, the single-channel conductance of expressed TRPC1 in kidney insect cell lines using whole-cell attach configurations of the patch clamp technique in the presence of 105 mM Ba^2+^ in the pipette solution was between 25 and 138 pS [104,105,107] (please see Table 1). However, the recording of expressed (in HSG cell lines) TRPP2 single channel showed a single-channel conductance in 2 mM Ba^2+^ between 1.2 and 4.5 pS [104,105,110,113] (Table 1). These differences in single-channel conductance levels make it challenging to identify the channel type. In addition, TRPC1 and TRPP2 channels were reported to be mainly present at the ER membrane and to act as calcium-released channels [110], causing their biophysical studies to be more challenging [108]. Furthermore, both types of TRPP channels had no naturally occurring activator or opener, making it difficult to study their presence, pharmacology, and contribution to intracellular homeostasis in physiology and diseases. In addition, the three reported blockers of these channels, amiloride, La^3+^, and Gd^3+^ [113] (please see Table 1) are not specific and block other types of Ca^2+^ channels, making it again difficult to study these types of channels. Furthermore, electrophysiological studies in HEK-transfected fetal human kidney cell line (HEK-293) showed that at −40 mV (according to the activation and inactivation curves), the activities of the naturally occurring TRPP2 take place at a membrane potential near 0 mV [117]. However, the resting membrane potential of these kidney cells is near –30 to −35 mV. Thus, since the channel’s activity is seen only at a membrane potential near 0 mV, this suggests that the TRPP2 channel is not active at resting membrane potential. Finally, according to single-channel studies, the activation of the TRPPC2 channel takes place during prolonged voltage depolarization [104,105,107], which suggests that TRP channels or, at least, TRPC1 and TRPP2 channels may have no inactivation gate, which is a characteristic of ROCs.

NMDAR and AMPAR channels are two subtype members of the ionotropic glutamate receptors (iGluRs) [102,109,124,125]. The NMDAR channel transports sodium and calcium inside the cells and potassium outside the cells [102]. Their exact single-channel conductance levels are unclear (please see Table 1). This problem could be due to using several ion species in the extracellular solution (Na^+^, K^+^, Ca^2+^, and Cl^−^) and the presence of Mg^2+^ in the pipette solution of the excised outside-out patch clamp configuration in HEK cells [125]. The recorded single-NMDAR-channel conductance levels were several: 23, 42, 61, and 89 pS (please see Table 1). These different conductance levels can be due to the solution used, which contained several types of ion species that NMDR channels can carry, such as Na^+^, K^+^, Ca^2+^, and Cl^−^ [126]. In addition, since Mg^2+^ blocks NMDAR channels, it is not recommended to use this ion in single-channel recording [124]. It is reported that this channel type seems to be mainly located in the central nervous system [127]. Their activation by their ligand glutamate contributes to learning and memory and neuronal migration [102]. Recently, this type of channel was reported to also be present in the cardiovascular system [102,109]. However, since the claimed effect on heart function is based on in vivo studies [109], it is possible that the cardiac effect could be secondary to the impact of NMDAR on the central nervous system (CNS), which should be verified in the future.

At the neuronal postsynaptic level, the AMPAR channel closely relates to the NMDAR channel [128,129]. Following glutamate binding to its receptor channel, an influx of Na^+^ depolarizes the membrane and removes Mg^2+^ from the NMDAR channel, which becomes permeable to Ca^2+^ and Na^+^ [128,129]. The influx of Ca^2+^ and Na^+^ through this channel further depolarizes the cell membrane (for review, see [102]). Compared to NMDAR channels, AMPAR channels are well characterized at the biophysical level [130,131]. AMPAR channels are the fastest iGluRs, and up to four subconductance levels (8.2, 18.8, 27, and 37 pS) have been reported [131] (Table 1). Recombinant AMPARs showed a single channel conductance of 7–8 pS [130]. The AMPAR conductance changes with unstable synaptic states [40] based on either an unlimited growth of synaptic strengths or a silencing of neuronal activity without additional homeostatic mechanisms [132]. As VOCCs, AMPAR channels induce calcium microdomains, resembling sparklets in neurons [18]. As mentioned earlier, it is challenging to study AMPARs in neurons since they work closely with NMDARs [129]. In addition, the regulation of the AMPAR channel conductance is very complex and is influenced by RNA editing and phosphorylation [40]. The postsynaptic localization of AMPAR provides these channels high importance in regulating synaptic transmission and cognitive function [133]. Whether all these characteristics of the AMPAR of the nervous system are shared by other cell types, such as those of the cardiovascular system, should be confirmed.

### 2.5. Sodium–Calcium Exchangers, Ca^2+^ Pumps and Ca^2+^ Release Channels

#### 2.5.1. Sodium–Calcium Exchangers

The sodium-calcium exchanger (NCX) uses the sodium gradient to drive calcium extrusion [134,135] (for more information, please see [136,137,138]). This exchanger is present in all membranes except for that of the ER/SR. The plasma membrane NCX has three isoforms, NCX1, NCX2, and NCX3, each localized differently throughout the body [139]. NCX1 undergoes alternative splicing, resulting in NCX1 with either exon A or B [136,140,141,142]. NCX1 with exon A is generally expressed in excitable tissues such as cardiomyocytes and neurons, whereas NCX1 with exon B is usually expressed in non-excitable tissues such as the pancreas and kidney [136,140,141,142].

NCX2 is usually expressed in the brain and spinal cord, while NCX3 is generally expressed in the brain and skeletal muscle [141,142,143,144] (Table 1). Therefore, the NCX1 isoform is present in the heart and is a plasma membrane protein capable of binding and transporting Na^+^ and Ca^2+^ across the plasma membrane [145,146]. Recent evidence also indicates its presence at the levels of mitochondria [119,120,121] and nuclear [16] membranes. It is a crucial Ca^2+^ efflux mechanism having an essential role in cytosolic, mitochondrial, and nuclear Ca^2+^ regulation. Thus, it may indirectly contribute to the regulation of the ER/SR function [147,148]. The NCX1 is a bidirectional ion transporter. Under normal physiological conditions, the NCX1 will exchange three extracellular Na^+^ ions for one intracellular Ca^2+^ ion. However, when the Na^+^ electrochemical gradient is reversed, such as in the case of membrane depolarization, the NCX will transport Ca^2+^ ions into the cell and Na^+^ ions out of the cell [140,145]. Following depolarization and contraction, the myocyte relaxation will begin once the Ca^2+^ is removed from the cytoplasm [149]. This can be partly accomplished by NCX1, which will pump out the calcium entered through the L-type calcium channels and restore the balance. Hence, NCX1 has an important role in the excitation–contraction coupling of the heart [149].

NCX1 is also implicated in certain pathologies as it prevents the accumulation of intracellular Na^+^, such as in ischemia/reperfusion and heart failure [150,151,152]. Therefore, if a pathology causes an intracellular Na^+^ overload, such as in cardiac hypertrophy [151], the NCX1 will extrude the excess Na^+^ and bring in Ca^2+^ [150,151,152]. This Ca^2+^ influx through the NCX1 induces an intracellular calcium overload, which promotes the activation of calcium-dependent kinases responsible, at least in part, for the development of cardiac hypertrophy [153]. Furthermore, many hypertrophic factors, such as angiotensin II and endothelin-1, activate the NCX1 [154,155].

NCX is also present at the inner mitochondrial membrane and is called Na^+^- and Li^+^-dependent mitochondrial Ca^2+^ release system that was later found to be linked to NCX and was called NCLX [156,157] (for review, please see refs. [158,159,160,161]). Similar to the ER/SR and the nucleus [8,11,16,17], the mitochondria are an essential intracellular reservoir of Ca^2+^. It contributes to regulating cytosolic Ca^2+^ homeostasis by removing intracellular Ca^2+^ overload and then releasing it slowly once intracellular free Ca^2+^ returns to an average level. The kinetics of intracellular Ca^2+^ uptake and release of free Ca^2+^ of the ER/SR and the nucleus is faster than that of the mitochondria. Similarly to the nucleus [8,11,16,17], the mitochondrial uptake and release of cytosolic Ca^2+^ from intracellular pools may depend on the minimum (100 nM) and maximum (500 nM) buffering capacities of the nucleus [7,8]. The proximity of the mitochondria to the ER/SR and the nucleus permits these three intracellular pools of Ca^2+^ of the uptake and release of Ca^2+^ to closely interact with buffer cytosolic Ca^2+^ overload and release this Ca^2+^ when needed to maintain normal resting cytosolic Ca^2+^ homeostasis, as shown in Figure 4. The close presence of the mitochondria around the nucleus and its high cytosolic free Ca^2+^ overload capacity prevents nucleoplasmic Ca^2+^ overload [7,8]. The prevention of nucleoplasmic free Ca^2+^ overload is also prevented by high densities of nuclear Ca^2+^ binding proteins, which act as a Ca^2+^ buffering system such as calmodulin.

The mitochondrial calcium uniporter (mtCU) is a Ca^2+^-gated ion channel complex that controls mitochondrial Ca^2+^ entry via the calcium-activated calcium channel ryanodine receptor type 1 present at the level of the mitochondria (mRyR1) [162,163]. For more details, please see refs. [162,164,165,166,167,168]. The mitochondrial Ca^2+^ uniporter regulator 1 (MCUR1) of the intermembrane space (IMS) carries Ca^2+^ influx into the mitochondria. Overexpression of MCUR1 promotes calcium uptake by the mitochondria [162]. The mtCU dysfunction plays an essential role in several diseases, such as cancer, Parkinson’s disease, and cardiovascular pathologies, including ischemia/reperfusion and pulmonary arterial hypertension [169,170,171,172,173]. Targeting the uniporter for the treatment of diseases related to the dysfunction of the mtCU has a significant translational potential for a novel therapy in particular cardiovascular diseases [168].

#### 2.5.2. Ca^2+^ Pumps

The exchange flickering between Ca^2+^ influx and efflux highly contributes to mitochondrial and ER/SR cytoplasmic free Ca^2+^ homeostasis. The direction of the Na^+^ efflux depends on the concentration of Na^+^ rise at the inner side of the exchanger [174]. In the absence of an increase in intracellular Na^+^ level, the exchanger will drive the efflux of Ca^2+^. Such calcium efflux is accompanied by efflux through the plasma membrane calcium Ca^2+^-ATPases (PMCA) and calcium influx into the ER/SR via a calcium pump called SERCA. The four basic PMCA isoforms (the PMCA is the product of four separate genes) have a tissue-specific expression [175]. The transcript of each of the four genes encoding PMCA pumps is subject to alternative splicing. The sites in which it occurs are named A, B, and C [175,176]. Of the many splice variants theoretically possible, about 30 have been detected at the RNA or protein levels [175,176]. PMCA1 and 4 are ubiquitously expressed and, thus, are thought to be the “housekeeping forms” [175,176]. PMCA2 and 3 are highly expressed in excitable cells and are known for their high basal activity, in contrast to PMCA1 and 4 for which the basal activity is much lower [175,176].

PMCAs were discovered by Schatzmann in 1966 [177]. Ten transmembrane domains form one, belonging to the P-type pump family [178,179,180,181,182]. The C-terminal cytosolic tail contains its binding sites for ATP and [183] calmodulin (CaM) [184]. PMCA has a high affinity and low capacity for calcium [175,185,186] and pumps Ca^2+^ from the cytosol against its electrochemical gradient. PMCA is stimulated by protein kinases A (PKA) [184], Ca^2+^/calmodulin kinase (CaMK), and is inhibited by protein kinase C (PKC) [175,187].

Ca^2+^-ATPases are also present at the ER/SR membrane [188] and are called the sarcoplasmic/endoplasmic reticulum Ca^2+^-ATPases (SERCAs) [189,190,191]. It is crucial in refiling the superficial and central ER/SR and decreasing cytosolic calcium overload. The SERCAs generate and maintain about a 10,000-fold Ca^2+^ gradient between the ER/SR lumen and the cytoplasm [187]. Three genes encode SERCA pumps (SERCA1-3), with SERCA2b and SERCA3 highly expressed in VSM [192,193]. SERCA is composed of 10 transmembrane (TM) domains [183] which include the two Ca^2+^ binding domains and three main cytosolic domains for ATP binding [182,191]. Phospholamban constitutes the intracellular active site of the pump, and its phosphorylation by PKA, PKG, and Ca^2+^-CaMK regulates the activities of the pump [182,191]. In addition, the sesquiterpene lactone, thapsigargin [194], is a specific blocker of this pump [189]. PMCAs and SERCAs were reported to be implicated in several cardiovascular diseases (for a review, see [152,195,196]). It is postulated that alterations in the function of the SR due to ischemia/reperfusion (R/R) depression in cardiac activity are accompanied by decreased SERCA and PMAC functions [152]. Thus, it is expected that there will be an increase in interest on the role of SERCAs in cardiovascular physiopathology as well as in developing a new strategy to treat pathologies related to defects in this type of transporter.

#### 2.5.3. Ca^2+^ Release Channels

The first report of the presence of an ER/SR-release channel appeared in the late seventies by Fabiato and Fabiato [197,198]. This calcium channel was sensitive to a plant with alkaloid ryanodine (Ry) [199]. This type of release channel was named the ryanodine receptor (RyR). It is present on the ER/SR [200,201] and is opened by calcium binding to the cytoplasmic side of the channel [202]. RyRs are located at the epicenter of the regulatory mechanism involved in excitation–contraction coupling [203]. Ryanodine binds with high affinity and selectivity to RyRs [201,204]. The channel assembles four RyR subunits of the same isoform (thus a homotetramer), forming a central Ca^2+^-conducting pore with a diameter of 2–3 Å [205]. Three genes encode RyRs, generating a specific isoform (RyR 1 to 3), with RyR 2 and 3 being the prominent isoforms [206,207]. RyR is anchored to the ER/SR membrane by interaction with the Ca^2+^ binding storage protein calsequestrin [208].

The second system that can release Ca^2+^ from the ER/SR pools is the inositol 3 phosphate sensitive receptor (IP_3_R) [209,210]. The IP_3_R is formed by co-assembling four subunits of about 300 kDa each [211,212]. Three genes code the IP_3_R subunits (isoforms IP_3_R 1 to 3) [211,212]. The receptor isoforms share 60 to 70% of amino acid residue homolog even though they differ in affinity for the ligand, inositol 1,4,5-inositol triphosphate (IP_3_), and their regulation by different modulators, such as Ca^2+^ [213,214,215], Mg^2+^, and ATP. Nevertheless, these isoforms share the same structural and functional organization; a ligand-binding domain, a large coupling (regulatory) domain that transduces the signal of ligand binding in addition to sites for Ca^2+^ and ATP binding, phosphorylation by protein kinases, and a short Ca^2+^ channel domain in the C-terminal portion [216]. IP_3_Rs form clusters of 25–35 receptors, with a cluster diameter of ~near 800 nm and a distance of several microns between clusters [217]. We must mention that cytosolic IP_3_ alone is insufficient to activate its receptor, and Ca^2+^ is required to activate the IP_3_R.

In summary, intracellular organelle calcium transporters play an essential role in simplifying the entry of calcium into the cytosol by releasing calcium via transporters such as ryanodine and IP_3_ sensitive channels and uptake of cytosolic calcium via Na–Ca^2+^ exchangers and Ca^2+^ pumps present at the nuclear, mitochondrial, and the ER/SR (SERCA) levels [8,16,17]. The proximity of these intracellular organelles to each other permits an efficient synchronization of the regulation of intracellular calcium homoeostasis. Finally, it is important to mention that mitochondrial Ca^2+^ overload contributes to reactive oxygen species (ROS) generation and apoptotic signaling [111,218], indicating the mitochondria’s importance in life and death.

### 2.6. Organic and Inorganic Calcium Channel Blockers

The most clinically used calcium blockers (or commonly called calcium antagonists) are directed against the cardiovascular VOCC L-type Ca^2+^ channels. The reason for this is historical [48] because of the discovery of the importance of calcium in contraction and cardiac action potential by the group of Coraboeuf in the early 60s [219]. At this time, the different subtypes of VOCCs were not known, and the channel was called the slow calcium channel in comparison to the fast kinetics of the sodium channel and was referred to as an inward slow current (Isi).

Calcium channel blockers have been used since their discovery in the seventies [220]. This Isi calcium blocker was used in several cardiovascular diseases, such as supraventricular tachycardia, hypertension, and angina pectoris.

In the 80s, Bean reported for the first time in atrial cells that Isi is the product of two types of VOCCs [221]. Later on, several papers appeared confirming this discovery in all excitable cells, such as ventricular cardiomyocytes [44,45]. At the same time, the first member of the dihydropyridine (DHP) family, nifedipine, was introduced as a calcium antagonist [58]. Similarly to nifedipine, several DHP derivatives, such as nitrendipine, nimodipine, nisoldipine, and isradipine (Figure 5) were introduced as L-type calcium blockers. As seen in Figure 5, isradipine (PN200-110) blocked the L-type calcium current (Figure 5A,B), and since DHP drugs are sensitive to light, turning on the light photobleached isradipine and restored the L-type current (Figure 5A,C). Turning off the light and reapplying isradipine inhibited once again the L-type current (Figure 5A,C). The DHP calcium blockers are known to act at the closing state of the channel and are thus used as antihypertensive drugs. In contrast, the phenylethylamine verapamil and its derivatives gallopamil (D600) and D888 act at the opening state of the L-type Ca^2+^ channel. It exerts an equipotent coronary vasodilation and negative chronotropic effect.

The benzothiazepine diltiazem is considered a weak calcium antagonist that shares both DHP and phenylethylamine effects of cardiac and vascular L-type calcium channels [57,222,223]. There are many other clinical applications for organic calcium channel blockers. Nimodipine is used to prevent neurological deficits, nifedipine in cerebral vasospasm, and all three categories of blockers found effective in Raynaud’s phenomenon [224]. One important aspect of organic L-type Ca^2+^ channel blockers is the stereo-selectivity of the drug (−)-enantiomers or (+)-enantiomers [50]. However, for example, the racemic form of verapamil is used clinically [50]. A critical aspect of the organic calcium blocker is their water and lipid solubility. As described in Section 3, all organic calcium blockers do not affect the L-type Ca^2+^ channel uniformly due to their specificity to specific subtypes of the channels (please see Table 1). Organic L-type Ca^2+^ channel blockers differ from each other depending on the gating mechanism of the subtype of the channel as well as its binding site in the aqueous pore and whether they decrease the probability of opening or decrease the duration of opening [225,226,227,228].

Inorganic VOCC blockers include Mn^2+^, Mg^2+^, Cd^2+^, Co^2+^, La^3+^, and zinc (Zn^2+^) [229]. Unlike organic Ca^2+^ blockers, inorganic Ca^2+^ blockers do not distinguish between the L-type calcium channel’s opening, closing, resting, and frequency. Therefore, they usually compete with Ca^2+^ to pass through the channel.

Organic ROCC blockers are a matter of extensive research due to the recent literature suggesting their implications in several cardiovascular and neuronal diseases (for review, please see refs. [61,101,102,109,124,125,130]). Among ROCCs, TRPC1 was reported to be the predominant isoform present in human-cultured endothelial cells [230]. As mentioned earlier, we limited our review to those well-studied TRPC1, TRPP2, NMDARs, and AMPARs. LOE-908 and amiloride block TRPC1 and TRPP2, as well as the inorganic blocker La^3+^ [231]. The AMPA receptor is blocked by the joro spider toxin (JSTX-3) and DNQX [101,130]. NMDA receptor is blocked by ifenprodil (selective GluN2B), D-2-amino-5-phosphonic-pentanoic acid (APS), MK-801, memantine, and by the inorganic blockers, Zn^2+^, Mg^2+^, and lead (Pb^2+^) [102,109]. There is no doubt that some ROCC blockers constitute an excellent future for clinical use, and particularly the NMDA receptor blocker memantine [232,233].

### 2.7. Role of Ca^2+^ Channels, Exchangers, and Pumps in Cardiovascular Diseases

It is not clear in the literature whether an increase in VOCCs’ densities and functions occur in cardiovascular diseases, which may contribute to an increase in blood pressure, cardiac arrhythmia, and hypertrophy. However, a decrease in the density and loss of α2δ-1 subunit of the L-type channels were reported [234,235]. This will logically tend to decrease blood pressure and the development of arrhythmia and hypertrophy. In addition, such a decrease in the density and loss of α2δ-1 subunit was reported to be compensated by an increase in the activities of the T-type Ca^2+^ channels [235] and/or TRPP2 [234,236,237] as well as TRPC1 [235,238].

An explanation of how L-type channels contribute to the development of cardiovascular diseases could be due to their stimulation by the signaling of receptors implicated in cardiac and vascular diseases such as those activated by: Ang II [1,3], ET-1 [11,17], neuropeptide Y (NPY) [3,109], and β-adrenergic agonists [44]. Hence, the overstimulation of L-type channels contributes to the development of hypertension [235,239,240], arrhythmia [241], hypertrophy [242], cardiomyopathy [243], angina [235], angiogenesis [14], and proliferation [12,244,245]. Thus, the beneficial effect of L-type Ca^2+^ channel antagonists is mainly due to the blockade of its overstimulation by an implication in cardiovascular diseases.

The abnormal functioning of L-type channels due to overstimulation increase in the signaling of receptors by second messengers generated by the activation of insulin receptors [234,236] was reported to take place in type 2 diabetes. This could be due in part to the increase in cytosolic and nuclear Ca^2+^ via insulin activation of R-type Ca^2+^ channels [7,11,92] and IP_3_ activating ER/SR Ca^2+^ release [92]. This may suggest that targeting the R-type Ca^2+^ channel and/or IP_3_ Ca^2+^ release channel could be used for the treatment of cardiovascular diseases in type 2 diabetes. This should be verified in the future.

It is also reported that other VOCCs, such as P-, Q-, and N-types, are implicated in neuronal dysfunction in diseases such as Parkinson’s [235,239,240], pain [235], epilepsy [235], Alzheimer’s [234], as well as obesity [235] and cancer [235]. The inhibition of N- and P/Q-type calcium channels by G-proteins contributes to presynaptic inhibition [84], which decreases ligand release and synaptic transmission. Furthermore, a decrease in the activity of the L-type Ca^2+^ along with ER/SR release affects excitation–secretion coupling, as well as the release of insulin from pancreatic cells [235].

Finally, due to the versatility and complexity of Ca^2+^ signaling [246] and its implication of excitation–contraction coupling as well as excitation–secretion coupling and excitation–gene transcription coupling, an abnormal functioning calcium homeostasis will contribute to the remodeling of many cell functions leading to cell proliferation and apoptosis. Thus, Ca^2+^ is life and becomes a killer when its intracellular homeostasis is compromised.

## 3. Discussion, Conclusions, and Perspective

Our knowledge of Ca^2+^ and Ca^2+^ transporters has evolved with the development of the technology to study the mechanisms responsible for their transport into and out of the living cell and its organelles. The discovery and commercialization of techniques, such as patch clamp and measurements of intracellular free calcium using fluorescent calcium probes and immunofluorescence coupled with 2D and 3D confocal microscopy imaging, have allowed the identification of the mechanisms and the intracellular sources of this ion. These different techniques, including that of the deletion or expression of different types of channels, have also permitted the identification of the subtypes of various transporters as well as their contribution to the functioning of excitable and non-excitable cells. Although the best method to determine the fingerprint of a Ca^2+^ channel is its unitary conductance, this method becomes slightly difficult when it comes to channels localized at the level of intracellular organelles such as the ER/SR, mitochondria, and nucleus. For this reason, the function and contribution of these channels, present at the membranes of the organelles, to cellular physiology and pathology are based on the interpretations of indirect evidence.

Furthermore, the controversy regarding the localization and role of some ROCC [111] types make it difficult to identify and study the function of specific channels, such as AMPA [40], NMDA [100,124,131], and TRP (TRPP2) [111]. In addition, the difficulty in studying the contribution of ROCCs in cardiovascular and neuronal physiopathology is also due to the lack of specific organic and inorganic inhibitors.

It is not clear in the literature whether the presence and the density of some types and subtypes of a Ca^2+^ transporter may be tissue- and sex-dependent [247]. Failure to consider these latter aspects can lead to different results from one laboratory to another, and thus fuel controversy and confusion in the literature.

Controversy can also arise from the expression of a transporter in cell lines that have nothing to do with normal cells, and most of the time, these originate from fetal tissues or cancer cells. Furthermore, expressing a transporter in a cell line will expose it to different environments than those in its native milieu, such as lipid composition and intracellular content. This latter aspect is crucial since almost all calcium transporters depend on the density of kinases involved in their activities. For all these reasons, great care must be taken when interpreting data obtained in cell lines.

In the 1960s, only the inorganic inhibitors of calcium transporters were available. These inorganic inhibitors act by binding to the transporter’s Ca^2+^ binding site. It was not until the 1970s, with the discovery of the first organic blocker, verapamil, that diseases related to plasma membrane Ca^2+^ transport were better understood and treated. The role of calcium channels and their pharmacology was made possible by the use of biophysical methods such as electrophysiology and, in particular, measurements of the cardiac action potential [57], vascular smooth muscle contraction [248], and endothelial cell secretion [249]. It is noteworthy that, since the discovery of the three classes of VOCC organic Ca^2+^ channel antagonists in the 1970s, there has been no advance in the development of new specific Ca^2+^ blockers for each subtype of L-type Ca^2+^ channels. Most notably, there has been the development of more DHP derivatives. The L-type DHP inhibitors have also been shown to affect the activity of other ion channels, including fast Na^+^ and T-type Ca^2+^ channels [250]. It is important to mention that some DHPs decrease contractility by inhibiting protein kinase C [251]. However, it is worth noting that L-type calcium channel blockers as well as other antihypertensive drugs do not affect resistant arterial hypertension [252], and the development of a specific blocker for such a type of hypertension awaits to be discovered.

On the other hand, calcium-permeable ROCCs are recently a subject for the development of specific organic blockers since they were reported to be implicated in neurodegenerative disorders (TRP and AMPA receptors) [100,125,133], mental health (AMPAR), neuropsychiatric disorders (NMDAR) [124], and cardiovascular diseases (NMDA and TRP receptors) [102,109,121]. Recently, the TRPM7 channel was reported to be implicated in pulmonary arterial hypertension [123]. Thus, it would not be surprising if other subtypes of ROCCs are involved in different cardiovascular diseases. Indeed, it is very important to increase our search for specific blockers of ROCCs in the future.

Similar to VOCCs and calcium-permeable ROCCs, NCXs are indirectly implicated in intracellular calcium overload in cardiac necrosis, hypertrophy, and heart failure in response to the increase in the activities of sodium–hydrogen exchangers [150]. Thus, any increase in intracellular sodium by any sodium transporter, including the fast sodium channel and taurine–sodium symporters, will indirectly induce calcium influx through the NCXs [253,254]. Although it is claimed that several cardiovascular diseases are related to NCXs, it is difficult to establish a direct implication because any up-regulation of the exchanger could be a rather adaptive mechanism to a cardiac pathology [175]. Thus, there is no report directly implicating NCXs in cardiovascular diseases. Amiloride and its derivatives block the exchangers but are not specific. However, SN-6, a new benzyloxy phenyl NCX, is considered the most specific inhibitor of the exchanger [255]. Therefore, it is possible to indirectly control the contribution of NCXs by targeting exchangers and symporters that induce intracellular sodium overloads, such as the sodium–hydrogen exchanger (NHE1) and the sodium–taurine cotransporter [140,150,151,253,254,256].

Both PMCA and SERCA pumps seem not to be directly implicated in cardiovascular diseases [175] since they mainly maintain normal resting intracellular Ca^2+^ homeostasis.

The abnormal functioning of calcium-released channels, such as IP_3_R and RyR types 1 and 2, are implicated in the development of cardiovascular diseases and neurodegeneration [257,258]. Therefore, several natural products were identified as specific blockers [257,258]. However, the presence of the Ca^2+^ release channels in the membranes of organelles makes it challenging to study and develop organic-specific blockers.

Finally, developing specific blockers for the different types and subtypes of Ca^2+^ transporters still awaits completion. In addition, further studies on the role of specific intracellular organelles, such as the nucleus and the mitochondria, in regulating cytosolic and nuclear homeostasis under normal and pathological conditions and developing specific antagonists of nuclear ion transporters remain to be accomplished. Targeting the disruption of mitochondria-to-cell redox communication represents a promising avenue for future therapy [259].

## Figures and Tables

**Figure 1 ijms-24-08803-f001:**
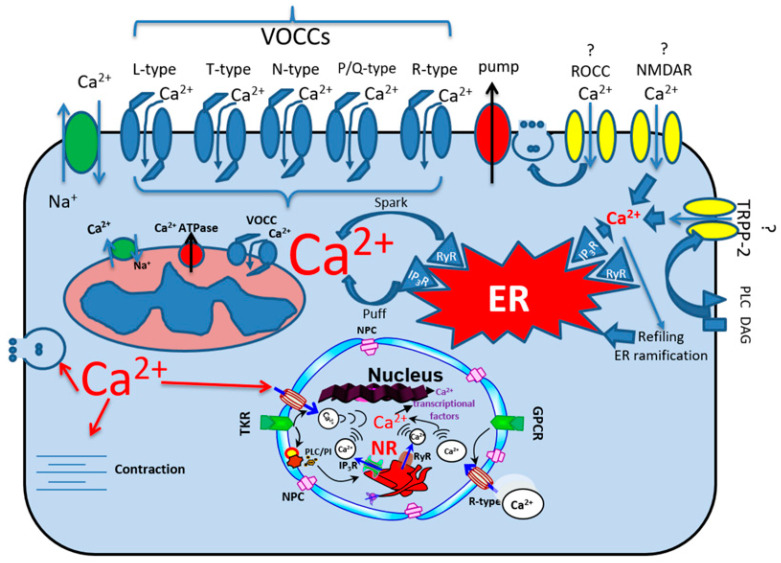
Schematic representation of various calcium transporters. VOCCs, voltage-operated calcium channel; ROCC, receptor-operated calcium channel; TRPP-2, transient receptor potential polycystic 2; PLC, phospholipase C; DAG, diacylglycerol; ER, endoplasmic reticulum; NR, nucleoplasmic reticulum; RyR, ryanodine receptor; IP_3_R, inositol 3 phosphate receptor; NPC, nuclear pore channel; GPCR, G-protein coupled receptor; TKR, tyrosine kinase receptor. Ca^2+^ in red represents the increase in calcium. The question mark (?) indicates that the nature of the single-channel conductance and the calcium selectivity of these ionic channels are unclear.

**Figure 2 ijms-24-08803-f002:**
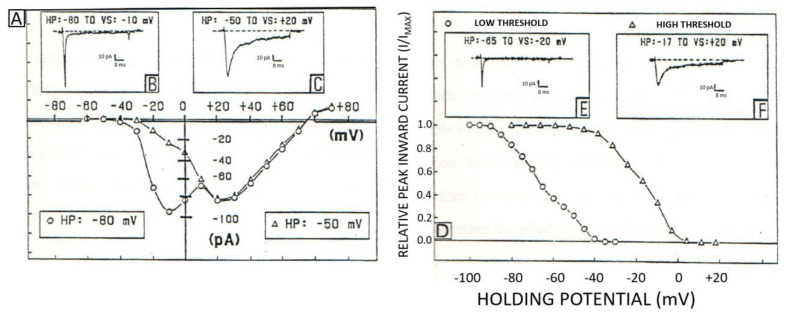
The whole-cell patch clamp technique shows examples of two types of Ca^2+^ currents in a human ventricular single cell. (**A**) Voltage-dependence of the low threshold of *I*_Ca_ (open circles) and separation of the high-threshold *I*_Ca_ (open triangles). (**B**) Peak current trace of the low-threshold *I*_Ca_ recorded from a holding potential (HP) of −80 mV with a voltage step (VS) to −10 mV. (**C**) Current trace of the high-threshold *I*_Ca_ was recorded from an HP of −50 mV with a VS to +20 mV. (**D**) Steady-state inactivation relationship of the low- (open circles) and high-threshold (open triangles) *I*_Ca_. (**E**) Low-threshold current recorded from HP of −65 mV with a VS to −20 mV. (**F**) High-threshold current trace recorded from HP of −17 mV with a VS to +20 mV. Currents were measured at the peak (modified from [44]).

**Figure 4 ijms-24-08803-f004:**
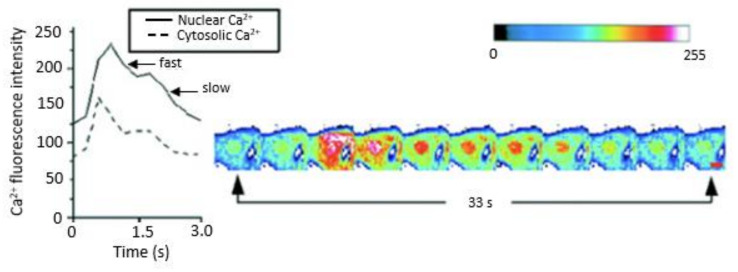
Using quantitative 3D confocal microscopy shows rapid time-lapse scans and graphic representations of cytosolic and nuclear free Ca^2+^ variations during spontaneous contraction in 10 day-old chick embryonic ventricular myocytes loaded with Ca^2+^ dye fluo-3. Whole-cell images show the relative fluorescence level and distribution of free Ca^2+^ during the propagation of Ca^2+^ waves. The graphic representations present the corresponding variations of the spontaneous waves of Ca^2+^ within the cytosol and the nucleus. The spontaneous wave of calcium quickly spreads across the cytosol and the nucleus. The free Ca^2+^ fluorescence intensity levels are particularly intense in the nuclear region and remain elevated even after cytosolic Ca^2+^ has returned to basal levels. The resting and peak levels of nuclear Ca^2+^ are higher than that of the cytosol. Spontaneous cytosolic and nuclear Ca^2+^ waves may have fast and slow decay components. Images are shown as pseudocolored representations according to the colored calibration bar of fluorescence intensity on a scale from 0 (black, absence of fluorescence) to 255 (white, maximal fluorescence) (modified from [17]).

**Figure 5 ijms-24-08803-f005:**
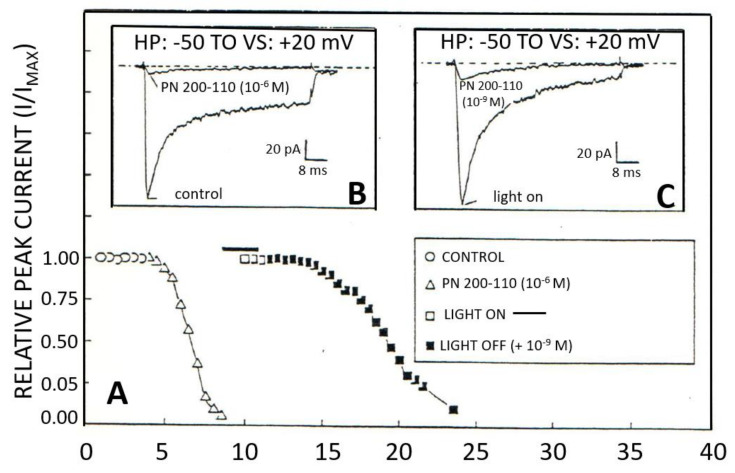
Using whole-cell patch clamp technique blockade of the L-type Ca^2+^ current in a human ventricular single cell by PN 200-110 (isradipine). The L-type *I*_Ca_ was activated from a holding potential (HP) of −50 mV with a voltage step (VS) to +20 mV. Superfusion with 10^−6^ M PN 200-110 completely blocked the L-type *I*_Ca_ within 5 min ((**A**), open triangles and (**B**), current traces in the left panel). Inactivation of PN 200-110 with a flash of light returned the L-type *I*_Ca_ amplitude to the control level ((**A**), open square and (**C**), right panel current trace). After turning off the light, a very low concentration of PN 200-110 (10^−9^ M) decreases the L-type *I*_Ca_ by 88% within 11 min (close square and current trace in right panel) (modified from [44]).

**Table 1 ijms-24-08803-t001:** Types and subtypes of calcium transporters and their conductance, localization, function, and organic and inorganic blockers.

Type	Subtype	Conductance (pS)	Localization	Function	Inorganic Antagonist	Antagonist Organic
VOCC-L	Cav1.1	25	SKELETAL MUSCLE	CONTRACTION	Ba^2+^, Mn^2+^, Ca^2+^, Mg^2+^	DHP, VERAPAMIL, DILTIAZEM
VOCC-L	Cav1.2	25	CARDIAC, VSM, NEURON	CONTRACTION, SECRETION, TRANSCRIPTION	Ba^2+^, Mn^2+^, Ca^2+^, Mg^2+^	VERAPAMIL HEART, DHPVSM
VOCC-L	Cav1.3	25	ENDOCRINE, NEURONAL, ATRIA, PACEMAKER	SECRETION, CONDUCTION	Ba^2+^, Mn^2+^, Ca^2+^, Mg^2+^	DHP, VERAPAMIL, DILTIAZEM
VOCC-L	Cav1.4	25	RETINAL	VISION, PHOTORECEPTOR	Ba^2+^, Mn^2+^, Ca^2+^, Mg^2+^	DHP
VOCC-P/Q	Cav2.1	10–20	NEURONAL	SECRETION		W-AGATOXIN
VOCC-N	Cav2.2	10–20	NEURONAL	SECRETION		W-CONOTOXIN
VOCC-R	Cav2.3	24	ALL CELL TYPES	RESTING [Ca^2+^]_i_	-	LOW CONCENTRATION OF ISRADIPINE
VOCC-T	Cav3.1	8–12	CARDIAC, NEURONAL	PACEMAKER	Ni^2+^, Ca^2+^	MIBEFRADIL
VOCC-T	Cav3.2	8–9	CARDIAC, NEURONAL	PACEMAKER	Ni^2+^, Ca^2+^	MIBEFRADIL
VOCC-T	Cav3.3	8–9	NEURONAL	PACEMAKER	Ni^2+^, Ca^2+^	MIBEFRADIL
AMPAR	-	8.2–37	ALZHEIMER, PARKINSON. DEPRESSION, EPILEPSY, CARDIAC, SKELETAL	MEMORY	-	PERAMPANEL TALAMPANEL JSTX-3 and DNQX
TRPC	1	25–138	RENAL,	REFILLING OF ER/SR STORE	La^3+^,Cd^3+^	SAR7334, SKF96365, MPEP
TRPP	2	80–160 1.2–4.5	RENAL, T-LYMPHOCYTE. VSM, VEC, ER/SR, CARDIAC	ENDOCRINE, PROLIFERATION, REFILLING ER, APOPTOSIS, SPERM FERTILISATION	La^3+^, Gd^3+^	AMILORIDE LOE-908
NMDAR	-	23–89	NEURONAL, CARDIAC, VEC	LEARNING, MEMORY, NEURONAL MIGRATION	Zn^2+^, Mg^2+^, and Pb^2+^	APS MK-801 IFENPRODIL MEMANTINE
Ca^2+^-PUMP	PMCA, SERCA	-	ALL CELLS	MAINTAIN Ca^2+^, HEMOSTASIS INTRACELLULAR	-	OUABAIN, THAPSIGARGIN
Na^+^-Ca^2+^ EXCHANGER	1	-	ALL CELLS	REGULATE INTRACELLULAR Ca^2+^ AND Na^+^ HOMEOSTASIS ER/SR Calcium	Li^+^	SN-6
RyR	1–3	-	ALL CELLS	RELEASE CHANNEL	-	DANTROLENE
IP_3_R	1–3	-	ALL CELLS	RELEASE CHANNEL	-	2-APB

VSM: Vascular smooth muscle, [Ca^2+^]_i_: intracellular calcium concentration; VEC: vascular endothelial cells.

## Data Availability

The data presented in this study are available upon request from the corresponding author.

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
