# Peer review of "Calcium Homeostasis, Transporters, and Blockers in Health and Diseases of the Cardiovascular System"

_ijms, 2023, doi:10.3390/ijms24108803_

Round 1

Reviewer 1 Report

“Calcium homeostasis, transporters, and blockers in health and diseases”

This review aims to describe the processes involved in calcium homeostasis from calcium influx and release from intracellular stores, and its extrusion or buffering. They also discuss a few blockers for calcium channels involved in these processes, and relate this to how they modulate the pathophysiology of various diseases.

General Comments:

This review aims to cover a large subject, but is rather patchy and lacks a comprehensive overview of the topic. The abstract sets a comprehensive tone that is not followed up in the manuscript itself.

Only a few calcium channels are discussed and therefore an explanation regarding why they focus on the ones they do should be included. For instance, there is no mention of calcium-permeable receptors (such as calcium-permeable AMPARs, NMDARs, etc.) – just calcium channels linked to upstream receptors (ROCCs). Only the sodium-calcium exchanger with relation to the mitochondria is discussed, but not the plasma membrane version. Calcium pumps which play important roles in homeostasis (such as the PMCA or SERCA) are also omitted. Furthermore, there is only a mention of two selected TRP channels (TRPC2 and TRPP) out of twenty eight (i.e. ignoring all the other TRPC members, TRPM, TRPV, etc.).

An overview of which cells are particularly vulnerable to disturbances in calcium homeostasis (neurons, cardiomyocytes,  …) should also be included, which would be important for understanding the relevance of calcium homeostasis to disease. In fact, the requirement for efficient maintenance of calcium homeostasis in cells per se is not explained (the single sentence on ln 40-42 is insufficient).

The language also needs some grammatical improvement. The point being made is often not clear, nor the relevance of things to the overall topic. Specifically, the subject of the sentence is not often clear (for example, ln 37 ‘In addition, its high-affinity binding’ presumably refers to calcium).

Specific points:

Section 1 - introduction:

The Introduction is rather confused – for example, ln 25 ‘all cell type’s embryogenesis, development, aging and death’ infers the embryogenesis of a cell whereas embryogenesis is a process involving multiple cell types. Do the author’s mean ‘all species embryogenesis, development, …’ or ‘all cell types involved in embryogenesis, …’?

Ln 38 - what is meant by a bound microdomain? How do microdomains contribute to intracellular release and uptake. This needs rewriting.

Ln 41 – it needs explaining how calcium transport dysregulation would affect ‘morphological and functional remodelling’. What does functional remodeling mean?

Section 2 – development of conduction and contraction systems in the heart

This section appears out of place with the rest of the manuscript with no clear connection. It does not mention calcium, homeostasis, channels, etc. Possibly it was intended to introduce a cell-type dependent upon calcium signaling, and sensitive to calcium disturbances in a disease setting (and therefore a potential target for medicinal calcium transporter targeting). However this point is not made, and it is not referred to again in the manuscript. The development of a chicken heart is not relevant to any of these potential points.

Section 3 – calcium ionic transporters

Ln 75 ‘ … highly contributes to positive charges’ density…’ needs rephrasing.

The nomenclature of VOCCs and ROCCs is inconsistent – with the clarification being on ln 85 ‘Voltage-gated calcium channels’ which would be VGCCs and RGCCs. In the legend to Figure 1, VOC is used (not VOCC) and explained as voltage-operated channel, etc. Please make this consistent.

In Figure 1 – Please explain what the question mark above ROCC and SOC refers to. There is a typo in the word refilling (it was spelt refiling, also in the Table).

Subsection 3.2 Please use a new paragraph for each different VOCC subtype which would make it easier to read and follow the logic. (i.e. ln 125 is a continuation of the previous paragraph; ln 168 – the T-type calcium channel is introduced mid-paragraph.

Ln 123 – ‘In some cell types, the L-type calcium channel is minimally or highly permeable to some ions’. Not enough information – which cell types, which ions?

Ln 159 – ‘The channel …’ it is not clear which channel is being discussed. Should be ‘This channel …’ if referring to N-types as introduced in the previous paragraph.

Table 1 – this is a list of an again unexplained selection of channels – receptor-operated channel ‘ROC’ should be expanded to include the relevant receptors. Similarly, store-operated channel ‘SOC’  involves many more channels than just TRPC2. ICRAC is not a channel – but refers to the current evoked by store-operated channels. As far as I am aware, amiloride targets TRPP3 and not TRPP2. Which pump does Ca2+-pump refer to – thapsigargin blocks the SERCA ATPase, whereas oubain targets Na+/K+ ATPase? What does Na+-Ca2+ refer to – presumably the sodium-calcium exchanger (but which one: NCX at the plasma membrane or NCLX at the mitochondria?) What does VSM stand for?

Figure 2 legend (ln 156) – is it necessary to explain that this recording is from cell no F0426? This information regarding cell number is omitted in the legend to Figure 4 and doesn’t contribute anything to the reader.

Subsection 3.3. It is not introduced what DHP-sensitive calcium channels are, nor why they have their own subsection. Why is the structure of these channels discussed but not the others?  The description is also unclear, how many subunits are there? What are their functions (is one the pore-forming subunit, and others playing auxiliary/regulatory roles)? Which subunits are referred to by (ln 194) ‘The Subunit (54kda) …’ or ‘The subunit (30Kda) …’?

Subsection 3.4. As mentioned before, there is no discussion of the calcium-permeable AMPARs or NMDARs. The explanation (ln 213) ‘These channels open in response to the activation of an associated receptor’ would suggest that they have only included calcium channels that do not directly bind a ligand, but are activated downstream of ligand-induced signaling. In fact, no examples of such a channel activated downstream of a ligand-binding event are actually given (just examples of Cl- channels and non-specific channels). The statement ‘Some receptor-associated ion channels are permeable to calcium’ needs to be supported – please give specific examples. Similarly, ‘Some calcium channels are regulated by G proteins while others are regulated by … reversible phosphorylation…’ - both need supportive examples.

Subsection 3.5. The first sentence contains a typo (ln 235 ‘are a members…’) and it is not clear whether it is meant that the TRP superfamily as a whole is well studied or only the specific TRP channels that have been selected for discussion. Also, please expand on what is meant by the ambiguity or controversy over their voltage-dependence. Ln 241 also contains typos – should read ‘Single channel recordings of native and expressed/overexpressed …’

The information regarding the conductance values are not given for any other channels discussed in the review, and their relevance is not explained.

Subsection 3.6. It reads as though mitochondria primarily function as reservoirs of calcium. It would be useful to put this into context – for example, explaining how mitochondrial activity is regulated by calcium. Similarly, the ER/SR function in not just as an organelle to buffer calcium, but also as a source of calcium to regulate cellular activity in response to cellular signaling. A fuller explanation of how the sodium-calcium exchanger can operate in forward and reverse depending upon the ionic gradient should be made.

Section 4 – calcium blockers.

This section gives only a very short overview of a few selected calcium channel blockers.

Also, please correct the references to (ln 318) calcium blockers or (ln 327) calcium antagonists to blockers or antagonists of the calcium ‘channel/transporters’.

Section 5 – roles of channels, exchangers and pumps in disease.

Again, only a very short overview of the topic. Essentially this is just a short summary of voltage-gated channel blockers – what about all the other calcium channels, exchangers or pumps (which are even in the title of this section)? From the title and abstract of the manuscript, much more would be expected here.

Ln 362 ‘Most drugs used to treat specific diseases are directed against … L-type calcium channels’. Is it meant that the majority of calcium-channel targeting drugs are directed against the L-type subtype, or really that of all available medicines, most of these are against this channel. Needs carefully rephrasing, and/or substantiating.

Please explain what the α2δ1 subunit (ln 373) is? Please explain how L-type dysfunction can activate R-type calcium channels (ln 376).

How do androgen and estrogen regulate calcium homeostasis? How are they related to diabetes and hypertension.

Author Response

Answer to reviewer 1 :

We thank the reviewer for his comments and suggestions. We did major revisions as suggested and as seen in the revised manuscript's copy. In addition, we tried, as much as possible, to make the revisions suggested.

Question 1: Only a few calcium channels are discussed, and there is no mention of AMPARs and NMDARs. Only the sodium-calcium exchanger concerning the mitochondria is discussed, and PMCA and SERCA are omitted. 

Answer 1: We included AMPARs and NMDARs and invited the readers to recent reviews concerning these ROCCs. We also expanded the section concerning the sodium-calcium exchanger, PMCA and SERCA.

Question 2: There is only mention of two TRP channels.

Answer 2: We did justify in the text why only two TRPs channels were chosen. As mentioned in the text, only channels whose biophysical characteristics were well-determined, and more particularly in the cardiovascular system, were covered in our review.  

Question 3: The language needs some grammatical improvement, and some sentences are not clear.

Answer 3: We did an in-depth revision of the manuscript. For this, we used highly sophisticated grammatical software, and one colleague and two of our Ph.D. students went through the manuscript

Question 4: The introduction is rather confusing.

Answer 4: We revised the introduction and deleted the section on embryogenesis and aging.   

Question 5: Previous line 38- what are microdomains.

Answer 5: For scientists working in calcium homeostasis, microdomains as well as nanodomains, are what we call sparklets, sparks, and puffs. This is now well-explained and documented.

Question 6: Line 41- It needs explaining how calcium transport dysfunction would affect morphological and functional remodeling.

Answer 6: We explained in more details this point.

Question 7: Section 2 appears out of place.

Answer 7: This section was deleted.     

Question 8: Section 3, previous line 75, needs rephrasing.

Answer 8: This was done.

Question 9: The nomenclature of VOCCS and ROCCS is inconsistent.

Answer 9: This was corrected throughout the manuscript and the figures. 

Question 10: Figure 1 – Please explain what the question mark and ROCC mean.

Answer 10: This was corrected as suggested.

Question 11: Previous section 3.2 use a new paragraph for each different VOCC subtype.

Answer 11: As suggested, the correction was done.

Question 12: Previous line 123- Not enough information, which cell type and ion.

Answer 12: This was made more clear.

Question 13: Table 1- Unexplained collection of channels. Amiloride targets TRPP3 and not TRPP2.

Answer 13: We revised Table 1 and explained in the text of the manuscript the different channels and transporters. Amiloride is one of the most non-specific compounds that block all ionic transporters, according to the literature and our experience. Thus, it is not surprising that it will also block TRPP3. 

Question 14: What calcium pump and sodium-calcium exchanger are referred to.

Answer 14: This is now well explained.

Question 15: Figure 2 legend: what is cell number.

Answer 15: This is now specified.

Question 16: Previous subsection 3.3- What DHP-sensitive calcium are?

Answer 16: It is now well-explained.

Question 17: Previous subsection 3.4- no discussion of AMPARS and NMDARs.

Answer 17: This is now covered, and references to G-proteins were removed.   

Question 18: Previous subsection 3.5- sentences contain typos and what is meant by ambiguity or controversy.

Answer 18: All the typos were corrected, and the sentence containing the words ambiguity or controversy was rewritten.

Question 19: The information regarding the conductance values are not given for any other channels discussed in the review, and their relevance is not explained.  

Answer 19: This was revised and made more clear. For a biophysicist that we are, the only way to really distinguish one channel from another, from a biophysical point of view, is single-channel conductance which is the fingerprint of the type of channel. This is now well-indicated. 

Question 20: Put in context the role of the mitochondria and the ER/SR.  

Answer 20: This was revised according to the reviewer's suggestion.

Question 21: Calcium blockers- Very short review of selected calcium channel blockers.

Answer 21: We added all known calcium blockers for all calcium transporters, as suggested.

Question 22: Role of channels, exchangers, and pumps in disease- Short review of the topic.

Answer 22: We expanded our review for all the transporters, and more particularly those selective for calcium.

Question 23: What does it mean: most drugs used to treat specific diseases are directed against… L-type calcium channels.  

Answer 23:  As suggested, we made this clear by indicating that the drugs are available in the clinic. We also rephrased previous lines 362, 373, and 376 and deleted the text related to androgen, estrogen, and diabetes.  

Reviewer 2 Report

Unfortunately, this review cannot be accepted for publication at this stage.

Comments:

I did not understand the inclusion of chapter 2 Development of conduction and contraction systems in the heart. What is the meaning of this chapter in the context of the review? This chapter looks completely out of place.

Figs 2 and 4. It is better to draw in a graphic editor, this will improve their quality.

Line 305. The mitochondrial calcium uniporter is located in the inner membrane, not the outer. In this case, it is necessary to specify in more detail about its structure and subunit composition, whose stoichiometry regulates the uptake of calcium.

Line 309. Mitochondrial NCX is called NCLX. It also requires a separate mention and more detailed disclosure. In this case, it is also worth noting the Ca2+-H+- exchanger and MPT-pore and their role in calcium transport.

In general, I would like to note that the authors provided very scarce information about calcium carriers, a great emphasis was placed on the channels of the plasma membrane, while the channels of organelles (ER and mitochondria) are presented very superficially. There is no information about SERCA, IP3R and other mechanisms. Although the title of the work implies a more extensive coverage of this topic. The authors should either consider the problem much deeper, or clarify the title of the work and correctly place the accents. Otherwise, the reader will not see a complete picture that reflects the work of channels in normal and pathological conditions.

A modern review should contain a significant number of references to recent sources. In this work, only about 15% of the sources are not older than 5 years, which is extremely small for such a recently popular topic.

Author Response

Answer to reviewer 2:

Question 1: I did not understand the inclusion of Chapter 2 (previous).

Answer 1: As suggested, we deleted this section.

Question 2: Figs 2 and 4- it is better to draw in a graphic editor. This will improve their quality. 

Answer 2: Unfortunately, figures 2 and 4 are published as unpublished results.

Question 3:Previous lines 305 and 309-  More details in NCX and NCLX.

Answer 3: As suggested, we made more clear the differences between NCX and NCLX.

Question 4: Should be clarified in more detail SERCA, IP3, and other mechanisms.

Answer 4: As suggested, we clarified in the abstract and in the introduction that we focus mainly our review on the cardiovascular system, particularly on calcium ionic transporters. We also, as suggested, gave more information on SERCA, IP3, RyR, and NCX.

Question 5: A modern review should contain a significant number of references to recent sources. In this work, only about 15% of the sources are not older than 5 years, which is extremely small for such a recently popular topic.

Answer 5: We added at least 50 references and referred the reader to recent reviews on the different subjects. Unfortunately, many recent reviews do not cite the origin of the information well and misinterpret the results described in older reviews by other authors. This is not ethical and jeopardizes the transmission of the correct information. Therefore, our review considers the original source of information, and as the saying goes: one must give Caesar what belongs to Caesar.

Round 2

Reviewer 1 Report

This is revised version is substantially improved, being more logical and clear.

However, there are still some corrections needed – including some points from the previous review which were not addressed.

For example, the introductory statement that ‘calcium plays a role in all cell types embryogenesis, development ..’ still remains. Embryogenesis and development are processes relating to an organism, not at the level of the cell.

Typos in Table 1 include refiling instead of refilling, intracellulaire instead of intra cellular, missing the word exchanger for the Na+-Ca2+ exchanger. What does EC (e.g. localisation of the NMDAR) refer to?

At the end of the legend for Figure 1 – there is no line break with the main text (ln 73) which is confusing.

Section 2 – lns 86-103 is missing references

Section 2.1 – introduction of the term receptor-operated calcium channels is not clear - the term ligand-gated channel would be preferable. However, if the authors wish to use this term, a clarification that this refers to the ligand-gated channels would be helpful – and examples should be given here, too. (Especially since in Table 1, the term ROCC is not used in reference to the channels listed there).

Furthermore, the authors state both in Section 2.1 (ln 112) and again in Section 2.3 (ln 252) that ROCCs are ‘not selective for a single type of ion’ – again in the title of Section 2.4 ‘nonselective ROCCs’. This is misleading since although these channels are considered non-selective for monovalent ions such as Na+ or K+, they display different permeabilities to calcium (i.e. with the net result that NMDARs and calcium-permeable AMPARs are more selective for calcium than sodium, whereas other AMPARs are more permeable to sodium than calcium). This should be clarified. Similarly, it is stated that ‘VOCCs are only selective for calcium’ (ln 124) but later the authors concede that the L-type channel is actually just ‘highly selective’ for calcium, and then list other ions to which this channel is permeable. This should be described more carefully.

The Figures 2, 3 and 4 need to be provided in better quality.

Ln 342 – please clarify what is meant by ‘unstable synaptic states’

Ln 344 – the sentence ‘It is also reported that AMPAR channels, as VOCCs induced calcium microdomains resembling sparklets in neurons’ needs rephrasing.

Ln 349-51 – It is unclear what is being compared with the nervous system when it is written ‘Whether all these characteristics of the AMPAR are similar to that of the nervous system’. Is the comparison AMPARs in the heart in comparison to that of AMPARs in the nervous system?

Ln 393 – By ‘As for the ER/SR ..’ is it meant ‘Similar to the ER/SR …’? and again in Ln 398, ‘As the nucleus …’ should read ‘Like the nucleus …’

Section 2.5.3 – lns 565-575. Only a few blockers are mentioned but not explained why – for example, it is not mentioned that NMDA receptors are blocked by MK-801 or memantine amongst others. Some explanation as to which blockers are clinically relevant would help.

Author Response

We thank the reviewer for his helpful comments. We did all the suggested revisions, and we added a new paragraph concerning  the R-type calcium channel as well as a new figure related to the biophysical characteristics of the channel. We also improved previous figures 2 and 4. We hope that all the revisions are satisfactory to the reviewer.

Question 1: For example, the introductory statement that ‘calcium plays a role in all cell types embryogenesis, development ..’ still remains. Embryogenesis and development are processes relating to an organism, not at the level of the cell.

Answer 1: We apologize that we forgot to do these revisions. They are now done.

Question 2: Typos in Table 1 include refiling instead of refilling, intracellulaire instead of intra cellular, missing the word exchanger for the Na+-Ca2+ exchanger. What does EC (e.g. localisation of the NMDAR) refer to?

Answer 2: We went through the table, and we made all the corrections, including those suggested by the reviewer. We also added new blockers for NMDAR and IP3R. We changed EC for VEC, and we indicated its meaning.    

Question 3: At the end of the legend for Figure 1 – there is no line break with the main text (ln 73), which is confusing.

Answer 3: This is done.

Question 4: Section 2 – lns 86-103 is missing references

Answer 4: For each sentence, we added a reference.

Question 5: Section 2.1 – introduction of the term receptor-operated calcium channels is not clear - the term ligand-gated channel would be preferable. However, if the authors wish to use this term, a clarification that this refers to the ligand-gated channels would be helpful – and examples should be given here, too. (Especially since in Table 1, the term ROCC is not used in reference to the channels listed there).

Answer 5: As suggested, we added that receptor-operated channels and ligand-gated channels mean the same thing.

Question 6: Furthermore, the authors state both in Section 2.1 (ln 112) and again in Section 2.3 (ln 252) that ROCCs are ‘not selective for a single type of ion’ – again in the title of Section 2.4 ‘nonselective ROCCs’. This is misleading since although these channels are considered non-selective for monovalent ions such as Na+ or K+, they display different permeabilities to calcium (i.e. with the net result that NMDARs and calcium-permeable AMPARs are more selective for calcium than sodium, whereas other AMPARs are more permeable to sodium than calcium). This should be clarified. Similarly, it is stated that ‘VOCCs are only selective for calcium’ (ln 124) but later the authors concede that the L-type channel is actually just ‘highly selective’ for calcium, and

Answer 6: As suggested, we made clear that in normal physiological solution, the L-type calcium channel is only selective for calcium; however ROCCs are not. There is not doubt that if we omit calcium from the solution, VOCCs with the exception of the R-type would carry a current such as Ba2+.

Question 7: Figures 2, 3, and 4 need to be provided in better quality.

Answer 7: As suggested, we did improve Figures 2 and 4. However, it was very difficult to improve previous figure 3 better.

Question 8: Ln 342 – please clarify what is meant by ‘unstable synaptic states’

Answer 8 : This is now done.   

Question 9 : Ln 344 – the sentence ‘It is also reported that AMPAR channels, as VOCCs induced calcium microdomains resembling sparklets in neurons’ needs rephrasing.

Answer 9 : As suggested, we rephrased the sentence.  

Question 10: Ln 349-51 – It is unclear what is being compared with the nervous system when it is written ‘Whether all these characteristics of the AMPAR are similar to that of the nervous system’. Is the comparison AMPARs in the heart in comparison to that of AMPARs in the nervous system?

Answer 10: We did the revision, as suggested

Question 11: Ln 393 – By ‘As for the ER/SR ..’ is it meant ‘Similar to the ER/SR …’? and again in Ln 398, ‘As the nucleus …’ should read ‘Like the nucleus

Answer 11: As suggested, this is now revised.

Question 12: Section 2.5.3 – lns 565-575. Only a few blockers are mentioned but not explained why – for example, it is not mentioned that NMDA receptors are blocked by MK-801 or memantine amongst others. Some explanation as to which blockers are clinically relevant would help.

Answer 12: As suggested, we added new blockers to the text and table. In addition, we added which blockers could be clinically transitional.

Reviewer 2 Report

Unfortunately, the authors edited the manuscript very superficially.

The authors write that they were citing classical sources. However, the problem is that there are currently hundreds of review papers on this topic. What is the novelty of this review? It is difficult to imagine that it consists in a new interpretation of classical data. Therefore, I emphasized that a modern review should discuss modern original papers, which are certainly based on classical data. I think the authors are wrong. This affects both the content and quality of the review. A couple of brief examples: the authors wrote one small paragraph about the role of the mitochondrial uniporter in calcium transport, this is a very popular topic now, so it is strange that the authors paid so little attention to it. The same applies to the phenomenon of calcium-dependent MPT pore, there is no information about it.

The authors write that «Unfortunately, figures 2 and 4 are published as unpublished results». But these are their own results, what is the problem to draw them in an acceptable way? The authors should show respect for the potential reader and show the figures in satisfactory quality.

Author Response

We thank the reviewer for his comments. We did all that was possible to be done.  

Question 1: The authors write that they were citing classical sources. However, the problem is that there are currently hundreds of review papers on this topic. What is the novelty of this review?

Answer 1: We already acknowledged the recent reviews in the field. For us, a review is not an original paper.

Question 2: It is difficult to imagine that it consists in a new interpretation of classical data. Therefore, I emphasized that a modern review should discuss modern original papers, which are certainly based on classical data. I think the authors are wrong. This affects both the content and quality of the review.

Answer 2: With all due respect to the reviewer, we do not agree with the comment. We simply have a different approach to writing a review than the reviewer.  

Question 3: A couple of brief examples: the authors wrote one small paragraph about the role of the mitochondrial uniporter in calcium transport, this is a very popular topic now, so it is strange that the authors paid so little attention to it. The same applies to the phenomenon of calcium-dependent MPT pore, there is no information about it.

Answer 3: The MPT is an old story, and being in the field, it is extremely difficult to study mitochondrial function and specifically the MPT in intact cells. Studying the mitochondria in isolated mitochondria is not physiologically accurate because, as we mentioned in our review, the functions of this organelle are closely related to the functions of the ER/SR and the nucleus. As suggested, we added few sentences concerning the MPT and invited the reader to very recent reviews in the subject.   

Question 4: The authors write that «Unfortunately, figures 2 and 4 are published as unpublished results». But these are their own results, what is the problem to draw them in an acceptable way? The authors should show respect for the potential reader and show the figures in satisfactory quality.

Answer 4: As suggested, we did our best to improve previous figures 2 and 4.